# Real-World Efficacy of Regdanvimab on Clinical Outcomes in Patients with Mild to Moderate COVID-19

**DOI:** 10.3390/jcm11051412

**Published:** 2022-03-04

**Authors:** Taeyun Kim, Dong-Hyun Joo, Seung Woo Lee, Jaejun Lee, Sang Jin Lee, Jihun Kang

**Affiliations:** 1Department of Internal Medicine, The Armed Forces Goyang Hospital, Goyang 10271, Korea; jimsb89@naver.com (T.K.); jdhyun2000@gmail.com (D.-H.J.); lsw-0518@hanmail.net (S.W.L.); pwln0516@gmail.com (J.L.); 2Department of Statistics, Pusan National University, Busan 46241, Korea; bir1031@hanmail.net; 3Department of Family Medicine, Kosin University Gospel Hospital, Busan 46241, Korea

**Keywords:** COVID-19, SARS-CoV-2, regdanvimab, remdesivir, dexamethasone, oxygen

## Abstract

Background: This study aims to evaluate the real-world effectiveness of regdanvimab on clinical outcomes in patients with mild to moderate coronavirus disease 2019 (COVID-19). Methods: This retrospective observational study included 152 patients (89 received regdanvimab and 63 did not) diagnosed with mild to moderate COVID-19 between August 2021 and October 2021 and admitted to Armed Forces Goyang Hospital. We collected information on the use of regdanvimab, remdesivir, dexamethasone, and supplemental oxygen; symptom severity score (SSS); and laboratory test results. A linear mixed-effects model was used to test the effectiveness of regdanvimab usage on SSS and the results of laboratory tests. A multivariate logistic regression model was used to calculate the odds ratio (OR) for additional therapeutic options, such as remdesivir, dexamethasone, and supplemental oxygen. Results: The patients who received regdanvimab were older, showed a higher rate of vaccination, and had a higher Charlson comorbidity index, initial body temperature, and percentages of pneumonia at admission. The use of regdanvimab showed no interactive effects on the SSS and laboratory findings. Older age, male sex, obesity, high initial body temperature, and the presence of pneumonia at admission were associated with increased ORs for the use of these additional treatments. The use of regdanvimab reduced the probability of requiring additional therapies such as remdesivir, dexamethasone, and oxygen supplementation by 90.3% (95% confidence interval (CI), 60.3–97.6), 85.8% (95% CI, 34.2–96.9), and 89.8% (95% CI, 48.3–98), respectively. Conclusions: Regdanvimab usage was well tolerated and was associated with a decreased probability of requiring remdesivir, dexamethasone, and oxygen therapy. However, changes in SSS were not significantly different by the drug usage.

## 1. Introduction

The spectrum of therapeutic options for COVID-19 is rapidly increasing. Since the Food and Drug Administration (FDA) and the European Medicines Agency (EMA) issued Emergency Use Authorizations for remdesivir in May 2020 and June 2020, respectively, several therapeutic options, including antiviral agents, monoclonal antibodies, and immune modulators, have been developed to improve outcomes in COVID-19 [1,2,3,4,5]. However, given remdesivir and dexamethasone are mainly used in hospitalized patients requiring supplemental oxygen [3,5], treatment options for patients with mild to moderate COVID-19 or those at high risk for disease progression are limited.

Regdanvimab is a monoclonal antibody that targets the surface spike protein of SARS-CoV-2 and has been shown to reduce hospitalization in patients with mild to moderate COVID-19 [1,6]. The EMA recommended the use of regdanvimab in patients with COVID-19 at a high risk of progression to severe disease in March 2021. In addition, the EMA’s Committee for Medicinal Products for Human Use issued a positive opinion for regdanvimab on 11 November 2021, and the European Commission authorized the marketing of regdanvimab on 12 November 2021. In September 2021, regdanvimab became the first treatment for COVID-19 approved by the Korean Ministry of Food and Drug Safety. However, there is still a paucity of data regarding its effectiveness and safety in clinical practice.

Therefore, the present study aimed to evaluate the real-world effectiveness of regdanvimab in patients with mild to moderate COVID-19 admitted to a single hospital dedicated to infectious diseases, compare outcomes in patients receiving regdanvimab with those in patients not administered regdanvimab, and investigate its safety profile and factors e treatment escalation.

## 2. Materials and Methods

### 2.1. Study Design and Participants

This was a retrospective observational study conducted at a single hospital that is dedicated to patients with mild to moderate COVID-19 on 25 August 2021 by the Korean Ministry of Health and Welfare (MOHW). The MOHW triaged patients with COVID-19 based on the disease severity, and our hospital was designated to manage patients with COVID-19 who were asymptomatic but at high risk, mildly symptomatic with slight limitation of activity, or those who needed oxygen supply through nasal prongs or facial masks.

Electronic medical records of all patients who were admitted to and discharged or transferred out of our hospital between August 2021 and October 2021 were reviewed. The diagnosis of COVID-19 was confirmed using real-time reverse transcription-polymerase chain reaction assays using samples from the upper or lower respiratory tract. Discharge and release from isolation were decided according to the following clinical criteria defined in the Response Guidelines of COVID-19 by the Korean MOHW (version 10) released on 17 May 2021: (1) at least 10 days have passed since symptom onset or diagnosis for asymptomatic patients and (2) at least 24 h have passed since the resolution of fever without antipyretic medication.

A total of 166 patients with mild to moderate COVID-19 were enrolled in this study. Among them, 14 patients were excluded for the following reasons: insufficient information on symptom score (*n* = 8) and the period from diagnosis or symptom onset to the time of admission >7 days (*n* = 6). Finally, 152 patients were included in the analysis; 89 patients had received regdanvimab, and 63 had not.

### 2.2. Data Measurements

#### 2.2.1. Baseline Characteristics and the Collected Information

The collected information included age, sex, height, weight, body mass index (BMI), vaccination history, initial vital signs (systolic and diastolic blood pressure, heart rate, respiratory rate, and the highest body temperature within the first 24 h of hospital admission), Charlson comorbidity index (CCI) score, symptom onset date, PCR testing date and result (cycle threshold values of targeted RNA-dependent RNA polymerase, and envelope and nucleocapsid genes), admission date, and discharge/transfer-out date. A patient was considered fully vaccinated ≥2 weeks after either single shot of Ad26.CoV2.S vaccine or second shot of a two-dose vaccine (ChAdOx1 nCoV-19 and BNT162b2 vaccine). The usage status and administration dates of regdanvimab, remdesivir, dexamethasone, and oxygen therapy were also collected. We reviewed the initial chest radiographs or computed tomography (CT) images to determine the presence of pneumonia at the time of admission.

Laboratory studies included white blood cell (µL), neutrophil (µL), lymphocyte (µL), hemoglobin (g/dL), and platelet (10^3^/µL) counts; C-reactive protein (mg/dL), blood urea nitrogen (mg/dL), creatinine (mg/dL), sodium (mmol/L), potassium (mmol/L), chloride (mmol/L), calcium (mg/dL), uric acid (mg/dL), albumin (g/dL), aspartate transaminase (IU/L), alanine transferase (IU/L), alkaline phosphatase (IU/L), lactate dehydrogenase (IU/L), gamma-glutamyl transferase (IU/L), and ferritin (ng/mL) levels; estimated glomerular filtration rate (mL/min/1.73 m^2^); activated partial thromboplastin time (s); and prothrombin time (s). The estimated glomerular filtration rate was calculated using the Modification of Diet in Renal Disease equation. All patients were tested on hospitalization days (HDs) 1, 4, and 7 as a part of routine clinical practice.

#### 2.2.2. Indication and Dosage of Regdanvimab

Regdanvimab received conditional marketing authorization in February 2021 and final approval in September 2021 from the Korean Ministry of Food and Drug Safety for the use in patients with mild to moderate COVID-19 who are at high risk of progression to severe COVID-19 [7]. The specific indications of the drug are as follows: (1) oxygen saturation > 94% without supplemental oxygen, (2) symptom onset ≤7 days prior to drug administration, and (3-1) BMI > 30 kg/m^2^, or (3-2) age > 50 years and at least one of the following underlying medical conditions: diabetes, hypertension, cardiovascular disease, chronic respiratory disease including asthma, chronic kidney disease, chronic liver disease, and immune-compromised state. Patients were administered the approved dose of regdanvimab—a single intravenous infusion of 40 mg/kg for 60 min. All patients in the regdanvimab group were administered the drug by HD 2. All patients who met the above criteria received regdanvimab during the entire study period. After the administration of regdanvimab, when patients required supplemental oxygen therapy, the clinician decided the next step of management (i.e., remdesivir or dexamethasone) according to the protocol used in previous studies [3,5].

#### 2.2.3. SSS

Because the interaction between physicians and patients was only through regular phone calls, the Committee of Infectious Disease of the hospital made a protocol for symptom surveillance on a regular basis. According to the protocol, all patients completed a self-administered questionnaire for SSS on HDs 1, 4, and 7. This symptom scoring system was designed to monitor COVID-19-related symptoms.

A paper questionnaire for monitoring COVID-19-related symptoms was designed using the guiding document from the United States Food and Drug Administration for assessment of COVID-19-related symptoms in outpatient adults and adolescents in clinical trials of drugs and biological products for COVID-19 prevention or treatment [8]. SSS was defined as the sum of individual scores of the following 14 COVID-19-related symptoms, with higher scores indicating severe symptoms: stuffy or runny nose (1–4), sore throat (1–4), shortness of breath (1–4), cough (1–4), tiredness (1–4), body ache (1–4), headache (1–4), chills (1–4), felling feverish (1–4), nausea (1–4), vomiting (1–4), diarrhea (1–4), dysgeusia (1–3), and anosmia (1–3).

### 2.3. Statistical Analysis

Study participants were categorized into the regdanvimab and non-regdanvimab groups. After checking the normality of the continuous variables, the *t*-test was used for parametric estimation and the Wilcoxon rank-sum test for non-parametric estimation. Categorical variables were compared using the chi-square test. The distributions of the SSS and laboratory results during the first, second, and third evaluations are descriptively presented using a boxplot graph.

A linear mixed-effects model (LMEM) was used to evaluate the safety of regdanvimab and to explore the interactive effect of regdanvimab on the SSS, considering the individual patient as a random effect factor and age, sex, BMI, CCI, and presence of pneumonia at admission as fixed effectors.

The effect of regdanvimab on the initiation of remdesivir, dexamethasone, and oxygen supplementation was evaluated using a multivariate logistic regression model after adjusting for potential clinical variables, such as age, sex, BMI, CCI score, vaccination status, initial body temperature, and pneumonia, at admission and the variables that showed significant differences between the regdanvimab and non-regdanvimab groups. Sensitivity analysis and variable selection were performed under the backward elimination process using the likelihood ratio test.

All statistical analyses were performed using R software, version 4.1.1 for Windows (R Development Core Team). Statistical significance was set at *p* < 0.05.

## 3. Results

Of the 152 patients with mild to moderate COVID-19, 89 (58.6%) had received regdanvimab, and 63 (41.4%) had not. Regdanvimab was administered on HD 1.7 (95% confidence interval (CI), 1.1–1.7) days. The regdanvimab group was older and had higher CCI scores, frequency of pneumonia at admission, and proportion of patients with complete vaccination than the non-regdanvimab group (Table 1). In addition, patients who received regdanvimab had higher baseline temperatures.

Regdanvimab decreased the probability of requiring additional treatments such as remdesivir, dexamethasone, and oxygen supplementation by 90.3% (95% CI, 60.3–97.6; *p* = 0.001), 85.8% (95% CI, 34.2–96.9; *p* = 0.013), and 89.8 (95% CI, 48.3–98; *p* = 0.006), respectively (Table 2). In addition, remdesivir usage was significantly associated with higher age, BMI ≥ 30 kg/m^2^, higher initial body temperature, and pneumonia at admission. Factors leading to dexamethasone usage were higher BMI, initial body temperature, and pneumonia at admission. In terms of oxygen supplementation, higher age, BMI, and initial body temperature; male sex; and pneumonia at admission were associated with therapy initiation. The findings from the sensitivity analysis were consistent with what was observed in Table 2, intensifying the probability of requiring additional treatments. In the sensitivity analysis, older age was associated with the initiation of remdesivir, oxygen therapy, and dexamethasone (Appendix A).

Safety profiles based on laboratory testing results using LMEM showed no significant differences between the regdanvimab and non-regdanvimab groups (Appendix A). However, in both groups, numerically decreasing trends were observed for C-reactive protein, creatinine, and lactate dehydrogenase. In addition, numerically increasing trends were observed for lymphocyte count, platelet count, activated partial thromboplastin time, and prothrombin time (Appendix A).

In terms of SSS, we observed no interaction between time and regdanvimab usage (Figure 1, Appendix A). Both respiratory and non-respiratory SSSs showed no association between drug usage and symptom improvement.

## 4. Discussion

The present study revealed that regdanvimab was well tolerated and was significantly associated with reduced probabilities of using additional therapeutic options, such as remdesivir, dexamethasone, and supplemental oxygen. In addition, obesity, higher initial body temperature, and pneumonia at admission were associated with an increased probability of using these treatments. Age and male sex showed a positive association with supplemental oxygen use; however, statistical significance was not reached. Although SSS gradually improved over 7 days, the degree of improvement was not different between the regdanvimab and non-regdanvimab groups.

A neutralizing monoclonal antibody against the S protein of SARS-CoV-2 has attracted attention based on preclinical data, indicating its potential antiviral effect in patients with COVID-19 [9]. The primary target of antibody therapies is the S protein that is responsible for binding to the host cell angiotensin-converting enzyme-2 (ACE2) receptor [10]. However, regdanvimab additionally binds to mutant proteins and interferes with the ACE2 receptor, suggesting its role in neutralizing potential SARS-CoV-2 variants [1]. Substantial viral clearance and alleviation of clinical symptoms were observed in regdanvimab-treated animals infected with the D614G variant [1].

In our study, regdanvimab was significantly associated with a decreased probability of using additional therapeutic treatments, such as remdesivir, dexamethasone, and oxygen supplementation. This finding is consistent with those of previous studies. In a phase 2/3 randomized, double-blind, placebo-controlled trial, regdanvimab was beneficial in reducing the need for hospitalization or oxygen supplementation [6]. The effect was more intense in patients with moderate disease (7.2% vs. 15.8%) and moderate disease plus age ≥50 years (8.8% vs. 23.7%) [6]. Our study showed that regdanvimab was associated not only with a decreased need for oxygen supplementation but also with reducing remdesivir and dexamethasone requirements by 90% and 85%, respectively. In another study, regdanvimab usage reduced death, oxygen supplementation, and the need for intensive care unit care by 83.1% (95% CI, 71.3–90.5) [11]. Although the proportion of patients with disease progression was slightly lower in the present study, the difference could be attributed to the heterogeneity in defining clinical outcomes and the younger age of patients receiving regdanvimab (46.9 vs. 61 years) in our study.

Other monoclonal antibodies for mild to moderate COVID-19 have been developed and show efficacy in reducing COVID-19-related medical visits, hospitalization, and death. The phase 3 BLAZE-1 study compared a single infusion of a combination of bamlanivimab plus etesevimab with placebo in patients with mild to moderate COVID-19, and only 2.3% of patients in the treatment group reported hospitalization or death compared to 7.2% in the placebo group [12]. In another phase 1–3 study, the REGN-COV2 antibody cocktail, which combines two monoclonal antibodies, casirivimab and imdevimab, reduced medically attended visits by 49% compared to the placebo group [4]. Sotrovimab, an engineered human monoclonal antibody, showed an efficacy by reducing the need for hospitalization or death due to any cause by 85% compared with placebo, with no identified safety concerns [13].

Obese patients (BMI > 30 kg/m^2^) had a much higher risk of receiving additional treatments than patients with a normal BMI. Our results are consistent with those of previous reports, concluding that obesity is a risk factor for disease progression in patients with COVID-19 [14,15]. Pneumonia at admission and a high initial body temperature were associated with an increased possibility of initiating remdesivir, dexamethasone, and oxygen therapy. The initial severity of pneumonia on CT could predict the clinical course in patients with COVID-19, with a more advanced grade indicating worse outcomes [16,17]. Although 50% of patients with COVID-19 may not have high body temperature at the initial presentation, the temperature may increase within a few days, and this clinical course might be a harbinger of worse clinical outcomes [18]. In addition, high body temperature may predict mortality in ventilated COVID-19 patients [19].

Male sex showed an increased tendency of the need for oxygen therapy, while the requirement of remdesivir or dexamethasone was not associated with the sex. Data from the OpenSAFELY cohort showed that men were more likely to experience severe illness than women [20]. Although the mechanisms underlying sex differences are unclear, women may show better innate and adaptive immune responses than men since the X chromosome contains the largest number of genes related to immunity in the whole human genome [21].

In terms of safety profile, the advisable system to report treatment-related events such as an adverse event reporting system was unfeasible, considering the retrospective study design. A previous study of six patients who received 40 mg/kg of regdanvimab reported adverse events in four patients with no grade 3 events [22]. In the BLAZE-1 trial involving outpatients with mild to moderate COVID-19, another virus-neutralizing monoclonal antibody, LY-CoV555, was evaluated, and no serious events occurred in the 309 patients in the treatment group [2]. In the present study, only laboratory markers were used to investigate safety by observing any interactions between regdanvimab administration and laboratory results. The LMEM showed no significant interactions. In addition, during admission, no serious adverse events related to regdanvimab usage, such as infusion reactions or anaphylaxis, were reported. Although the application of LMEM to evaluate safety profiles may not be appropriate and may warrant further well-designed clinical trials, this result might provide insights into the safety profile of regdanvimab.

A previous clinical trial reported that regdanvimab had a beneficial effect on symptom resolution in COVID-19 [6]. However, the present study did not show any significant difference in SSS between the regdanvimab and non-regdanvimab groups. In addition, when we categorized symptoms into respiratory and non-respiratory symptoms, there was no difference between the two groups. One possible reason for these discrepancies could be the difference in the symptom scoring system between the studies. We used SSS, a robust symptom scoring scale, to quantify dynamic symptom changes over time; however, other studies on regdanvimab classified COVID-19 symptoms in a less systematic way. Further studies are needed to objectively quantify and evaluate symptom changes in patients receiving regdanvimab.

Our study had several limitations. First, considering that this was a retrospective observational study, unmeasured confounders may have impacted the observed findings. Although we robustly adjusted expected, important confounders such as vaccination status, other variables might have affected the results. Second, the safety profile was indirectly investigated using laboratory samples, and an adverse event reporting system was not used. Third, because the study participants were mainly included patients with mild to moderate COVID-19, specific markers used to judge treatment efficacies, such as mortality, use of mechanical ventilation, and length of intensive care unit stay, could not be evaluated. Fourth, we could not study the impact of regdanvimab on the duration of hospital admission because all patients remained admitted to the hospital for a predefined duration, in line with the national isolation guideline.

Despite these limitations, we expect that the current study has important clinical implications in the real practical field. However, evaluating which patients would respond to regdanvimab may be of interest, and further studies to introduce a prediction model or construct a cluster based on machine learning might be useful. In addition, although identification of variants was unfeasible in the present study, given that delta variant is at the time of this writing (November 2021) identified over 90% of new cases of COVID-19 in South Korea, many patients infected with the delta variant would have been included in the analysis, and our result may implicate clinical effectiveness of regdanvimab against this variant.

## 5. Conclusions

Regdanvimab is well-tolerated and may be beneficial in reducing the need for additional therapeutic options such as remdesivir, dexamethasone, and supplemental oxygen, although symptom alleviation was not associated with regdanvimab use. Further studies with larger samples are warranted, and a prediction model or clustering algorithm may provide insights into predicting who would respond to the treatment.

## Figures and Tables

**Figure 1 jcm-11-01412-f001:**
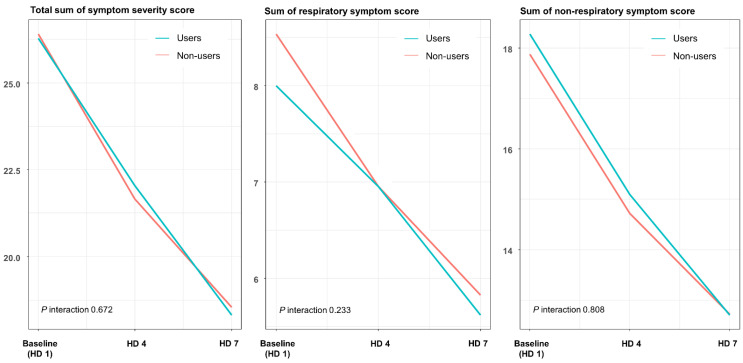
Estimated symptom severity scores of respiratory and non-respiratory symptoms according to the usage of regdanvimab using a linear mixed-effect model. HD = hospitalization day.

**Table 1 jcm-11-01412-t001:** Clinical characteristics according to regdanvimab usage.

	Users (*n* = 89)	Non-Users (*n* = 63)	*p*-Value
Age	46.9 (43.9–49.8)	36.1 (32.9–39.3)	<0.001
Men	44 (49.4)	33 (52.4)	0.721
BMI	27.0 (25.9–28.2)	25.3 (24.0–26.7)	0.063
PCR result (Ct value)			
RdRp	17.8 (16.54–19.06)	18.31 (16.79–19.82)	0.611
Envelope (E) gene	19.05 (17.83–20.23)	19.12 (17.53–20.7)	0.946
Nucleocapsid (N) gene	16.12 (14.29–17.95)	16.69 (14.65–18.73)	0.679
Charlson comorbidity index	0.9 (0.67–1.13)	0.3 (0.12–0.48)	<0.001
Vaccine completion			0.008
No	67 (75.3)	58 (92.1)	
Yes	22 (24.7)	5 (7.9)	
Smoking			0.643
Never	61 (68.5)	43 (68.3)	
Former	4 (4.5)	5 (7.9)	
Current	24 (27.0)	15 (23.8)	
Initial vital sign			
Systolic BP	124.4 (121.3–127.4)	121.3 (118.3–124.4)	0.267
Diastolic BP	81.7 (79.6–83.7)	80.6 (78.1–83)	0.475
Heart rate	82 (79–85)	80 (77–83)	0.476
Respiratory rate	16.1 (15.7–16.5)	16 (15.6-16.4)	0.835
Body temperature	37.9 (37.7–38.1)	37.4 (37.3–37.6)	<0.001
Length of admission	9.5 (9.2–9.9)	9.1 (8.8–9.4)	0.111
Transfer-out or death	2 (2.2)	0 (0)	0.511
Pneumonia at admission	58 (65.2)	15 (23.8)	<0.001
Remdesivir use	23 (25.8)	14 (22.2)	0.608
Dexamethasone use	9 (10.1)	7 (11.1)	0.843
Oxygen supplement	12 (13.5)	9 (14.3)	0.888

Data are presented as mean and 95% confidence interval for continuous variables and numbers and percentages for categorical variables, otherwise stated. For continuous variables with normal distribution, a *t*-test was performed, and for variables that violated the normality, a Wilcoxon rank-sum test was performed (SBP, DBP, and HR). BMI = body mass index; PCR = polymerase chain reaction; Ct = cycle threshold; RdRp = RNA-dependent RNA polymerase; BP = blood pressure.

**Table 2 jcm-11-01412-t002:** Impact of regdanvimab use on the initiation of remdesivir, dexamethasone, and oxygen in patients with mild-to-moderate COVID-19.

	Remdesivir			Dexamethasone			Oxygen Supplement		
	AOR	95% CI	*p*-Value	AOR	95% CI	*p*-Value	AOR	95% CI	*p*-Value
Regdanvimab									
Users	0.097	0.024–0.397	0.001	0.142	0.031–0.658	0.013	0.102	0.02–0.517	0.006
Non-users	1			1			1		
Age, years	1.104	1.031–1.182	0.005	1.063	0.977–1.157	0.157	1.083	0.998–1.175	0.056
Sex									
Women	1			1			1		
Men	1.965	0.697–5.545	0.202	1.544	0.373–6.388	0.549	4.597	0.974–21.686	0.054
BMI, kg/m^2^									
<25	1			1			1		
25–<30	2.527	0.748–8.539	0.136	7.955	1.182–53.527	0.033	4.173	0.782–22.279	0.095
≥30	5.838	1.414–24.092	0.015	12.105	1.534–95.553	0.018	9.905	1.416–69.311	0.021
Charlson comorbidity index	0.705	0.308–1.613	0.407	0.92	0.272–3.108	0.893	1.442	0.509–4.087	0.491
Vaccinated									
Fully	1			1			1		
Not fully	0.66	0.13–3.362	0.617	0.68	0.079–5.873	0.726	0.74	0.098–5.575	0.77
Initial body temperature	4.445	1.981–9.973	<0.001	2.666	1.034–6.875	0.043	2.942	1.162–7.45	0.023
Pneumonia at admission									
No	1			1			1		
Yes	9.923	2.657–37.06	0.001	19.071	1.703–213.515	0.017	24.096	2.47–235.047	0.006

AOR was calculated after adjustment for age, sex, BMI, charlson comorbidity index, vaccination, initial body temperature, and the presence of pneumonia at admission. AOR = adjusted odds ratio; CI = confidence interval; BMI = body mass index.

## Data Availability

The data that support the findings of this study are available from the corresponding author upon reasonable request.

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
