# Peer review of "Real-World Efficacy of Regdanvimab on Clinical Outcomes in Patients with Mild to Moderate COVID-19"

_jcm, 2022, doi:10.3390/jcm11051412_

Round 1

Reviewer 1 Report

This is an interesting study, with a well defined methodology, to assess the real word effectiveness of regdanvimab, a neutralizing antibody against SARS-CoV-2. The article is well written, the data is presented clearly and potential study limitations are well discussed, in order to allow the readers to better understand the degree to which the data can be generalizable to other patient populations.

I have listed below a list of comments and queries that I feel would make the manuscript clearer and easier to read and understand for the journal’s readership:

  • I would suggest that the authors add to the title the fact that this a real-world effectiveness study.
  • Also, throughout the manuscript, it would be better to replace “efficacy” (which is shown in clinical trials) with “effectiveness” (which is confirmed through clinical practice) – lines 17, 55, 282.
  • I would suggest replacing CT-P59 with regdanvimab throughout the manuscript, for consistency with its approved name.
  • Line 20: Please specify in which hospital the study was performed.
  • Line 22: Please replace the term “interactive”
  • Line 23: This should be laboratory test “results”.
  • Lines 25-27: This phrase is very vague. Please specify how they were different. These are baseline characteristics, and therefore it would be clearer to state that patients who received regdanvimab were older, had higher Charlson comorbidity index scores, etc.
  • Line 27: What does “time of survey” refer to?
  • I would suggest moving the phrase on lines 30-32 (Older age…) before the phrase starting on line 28 (The use of CT-P59…).
  • Line 33: I would suggest changing “using” with “requiring”.
  • Lines 41-43: Please use the plural for “agents”, “antibodies”, “modulators”.
  • Line 47: The term “is expected” can be replaced with “has been shown” and the reference #10 from your reference list could be moved here, as it is the pivotal phase 2 trial showing efficacy of regdanvimab.
  • Line 58: I would suggest replacing “affecting” with “leading to”.
  • Line 74: If these guidelines were published on 31 August, was another guideline active throughout the month of August (which is part of the study period)?
  • Line 81: Please revise to “had received” and “had not” to make it clear that this had already happened in clinical practice.
  • Line 81: Please also specify here that this had happened part of routine clinical practice.
  • Line 86: The first 24 h of disease or of hospital admission?
  • Line 103: As part of routine clinical practice?
  • Line 113: This is probably not moderate “symptoms” but “moderate COVID-19”, i.e., patients with pneumonia, right?
  • Line 108: Please state when regdanvimab started to be used in clinical practice in Korea (this is relevant to better understand why the study period August-October was chosen)
  • Line 114: Please specify who took the decision whether to treat with regdanvimab or not). It would be important to understand whether in routine clinical practice all patients meeting the risk criteria described here received regdanvimab during the whole study period.
  • Line 115: Please define “HD”. Is it hospitalization day?
  • Line 129: Could “febrile sense” be rephrased to “feeling feverish”?
  • Line 132: For continuous variables was distribution checked? The t test is only appropriate for variables with parametric distribution.
  • Line 137: Please define what “time of survey” refers to.
  • Lines 152-153: This phrase is not clear. Are you referring to the fact that patients who received regdanvimab had higher baseline temperature?
  • Table 1: Please rephrase “users” and “non-users”. It might be better to rename them as “non-regdanvimab” and “regdanvimab” groups.
  • Table 1: Vaccine completion – it is not clear why “yes” and “no” do not add up to 100%? 92.1+11.2=103.3. And 73.2+15.9=89.1
  • Line 164: Please revise “affecting”
  • Table 2, Figure 1: Please rephrase “users” and “non-users”.
  • Figure S1, S2, S3: Please rephrase “users” and “non-users”. Please specify on the X axis the day of hospitalization corresponding to each evaluation.
  • Tables S1 and S2: Please rephrase “users” and “non-users”. For non-users, please revise “before” to “baseline” in the column name. Please specify the day of hospitalization corresponding to each evaluation. Please explain in a table legend what P”tint” refers to. Please revise uL to µ
  • Figure 1: On the X axis, please also specify the number of hospitalization days.
  • Line 187: Do you mean “higher” initial body temperature?
  • Line 200: Please specify which variants were testes in animals.
  • Line 201: Please specify: “in our study”.
  • Line 205: I am not sure where in the cited reference (#10) you found the quoted percent, of 54%. The data from part 1, which is cited in reference #10 was also presented at ECCMID 2021 and the percentage was 66.1% in high risk patients for part one of the study, as reported in the abstract available here: https://doi.org/10.1093/ofid/ofab466.745. In the same ECCMID presentation, on the slides, data from part 2 of the study was presented, with 70% reduction rate in all patients and 72% reduction rate in high-risk patients.
  • Also, I would suggest updating reference #10, which is cited as a preprint, with the accepted version of the same paper, accepted for publication in Open Forum Infectious Diseases in 2022.
  • Line 206: Please revise: “and age ≥50 years (7.5% vs. 23.7%)” to “and moderate disease plus age ≥50 years (7.5% vs. 23.7%).” Also, after this phrase, the same reference (#10) should be cited again, to explain where these percentages came from.
  • Line 207: Please revise to “decreased need for oxygen supplementation”.
  • Line 208: Please revise “use” to “requirement”.
  • Line 237: This phrase is not very clear.
  • Line 274: Please add a limitation regarding the fact that you could not study the impact of regdanvimab on duration of hospital admission because all patients remained admitted to the hospital for a predefined duration in line with the national isolation guideline.
  • Line 279: Instead of “now” please specify “at the time of this writing” and specify the month and year when you made this statement. This is important as omicron has started to replace delta in many countries.
  • Line 286: Please replace “drug use” with “regdanvimab use”.

Reviewer 2 Report

Taeyun Kim  et al report clinical outcomes in a retrospective, monocentric cohort  of 89 patients with mild to moderate COVID-19 treated with regdanvimab, compared with 63 patients who did not receive the treatment. Despite the obvious limits inherent in this study design, which are appropriately acknowledged in the discussion, this is an interesting case-series reporting real-world efficacy of monoclonal antibodies in COVID-19.

However, I have a few concerns:

- the main study outcomes are all based on treatment escalations (use of remdesivir/ dexamethasone/ oxygen). The treatment protocols and the criteria for escalation should be clearly described in the methods

- the main analysis consists of logistic regression models using 8 predictors. Given the low number of events (max n=37 events for remdesivir use), using such a high number of predictors may affect  the reliability of results. Ideally, increasing the sample size could strengthen the results. At a minimum, it would be appropriate to present a sensitivity analysis with the results of models from which weaker predictors have been excluded.

- It is stated that the treatment group had higher proportion of patients with incomplete vaccination than controls (line 151). However, this does not seem in line with the numbers presented in table 1, please clarify

- The percentages in table 1 do not seem in line with the absolute numbers reported. For instance, the percentage of men among users is 45.1, but 44/89=49.4%

Round 2

Reviewer 2 Report

I am happy with the changes that the Authors made according to my comments.

I have identified only a few more minor points to be addressed:

- line 26: “The patients who received regdanvimab were older, showed a lower rate of vaccination”. This is not in line with table 1, please clarify

- lines 70-73: “This was a retrospective observational study conducted at a single dedicated hospital for patients with mild to moderate COVID-19 by the Korean Ministry of Health and Welfare (MOHW) on August 25, 2021”. This sentence is not clear, was the study carried out in one day? Please clarify

- line 122: please define what is exactly meant by “moderate COVID-19“

- line 167-168: “The 152 patients with mild to moderate COVID-19, 89 (58.6%) received regdanvimab 167 and 63 (41.4%) did not.“ The grammar of this sentence is not correct, please rephrase

-line 190: please specify how glomerular filtration rate was estimated

- lines 188-192: This passage is not clear at all, please rephrase.